# Temperature Distribution of Vessel Tissue by High Frequency Electric Welding with Combination Optical Measure and Simulation

**DOI:** 10.3390/bios12040209

**Published:** 2022-03-31

**Authors:** Hao Wang, Xingjian Yang, Naerzhuoli Madeniyeti, Jian Qiu, Caihui Zhu, Li Yin, Kefu Liu

**Affiliations:** 1Academy for Engineering & Technology, Fudan University, 220 Handan Road, Shanghai 200433, China; 19210860056@fudan.edu.cn (H.W.); 19210860011@fudan.edu.cn (X.Y.); 20210860101@fudan.edu.cn (N.M.); 2School of Information Science and Engineering, Fudan University, 220 Handan Road, Shanghai 200433, China; 20110720029@fudan.edu.cn (C.Z.); 19210720016@fudan.edu.cn (L.Y.); jqiu@fudan.edu.cn (J.Q.)

**Keywords:** electrosurgery, tissue welding, temperature distribution, Raman spectrum

## Abstract

In clinical surgery, high frequency electric welding is routinely utilized to seal and fuse soft tissues. This procedure denatures collagen by electrothermal coupling, resulting in the formation of new molecular crosslinks. It is critical to understand the temperature distribution and collagen structure changes during welding in order to prevent thermal damage caused by heat generated during welding. In this study, a method combining optical measurement and simulation was presented to evaluate the temperature distribution of vascular tissue during welding, with a fitting degree larger than 97% between simulation findings and measured data. Integrating temperature distribution data, strength test data, and Raman spectrum data, it is discovered that optimal parameters exist in the welding process that may effectively prevent thermal damage while assuring welding strength.

## 1. Introduction

Over the past few decades, high frequency electric welding (H.F.E.W.) can play an increasingly important role in the field of surgery, such as in the excision of ulcers [1], colorectal surgery [2,3], ablation of tumors [4,5], and sealing vessels [6,7,8], by delivering a high-frequency current. The surgical instruments are used to seal vessels by clamping them down and passing a high-frequency current through the tissues. The collagen in vessel tissue denatures to form a firm weld area as a result of the synergistic action of current and resistance heat generated by the vessel in the preceding process, completing the fixation and sealing between tissues. Because there are no foreign bodies, H.F.E.W. heals faster and has less of an impact than needle sutures [3,9].

However, in addition to the above advantages, the use of H.F.E.W. will inevitably cause heat to accumulate in vessel tissue, which may affect the therapeutic effect and even lead to irreversible thermal damage [3]. This problem urges people to better understand the temperature distribution in vascular tissue when using H.F.E.W.

There are many ways to measure temperature change in the tissue welding process at the moment, but they are limited by the electrode structure, so the electrode must be modified. Yang C.H. [10] embeds the thermocouple on the insulating resin base, measures the thermal conductivity and cooling process after welding, and reversely deduces the temperature rise change during welding. Cezo J.D. [11] embedded a set of thermocouples in the device and tissue, measuring the temperature change data of seven points in the two directions of the *x*-axis and *y*-axis on the welding plane, respectively; Chen R.K. [12] uses the method of FEM model simulation analysis of the welding process. These studies have observed and analyzed the temperature change in the welding area to indicate the temperature distribution and the changes in the physicochemical properties of tissue, which is very important for improving welding strength and reducing incidental thermal damage, but further optimization experiments are still needed.

The heat change shall be directly observed rather than speculated on the basis of data obtained after welding. The accuracy of temperature measurements in the thermal diffusion area should be improved, and the sample size should be increased. It is very important for deducing the temperature distribution data of the welding zone. To simulate the actual operation environment, the surgical instruments should not be changed as much as possible.

Considering the aforementioned insight, we present a new temperature distribution measurement method that combines infrared thermal imager observations with modeling. On the precondition that the electrode would not be reformatted, the temperature change of vessel tissue (electrode uncovered) was monitored, and the temperature distribution of the tissue center region (electrode coverage area) was estimated based on simulation data analysis.

We perform welding strength testing and Raman spectroscopy on the welded tissue. From the data obtained from the weld strength test, we found that as the tissue temperature rises, the weld strength also increases, but after exceeding the threshold, the welding success rate is reduced. Raman spectroscopy data also show that during the welding process, collagen is “destroyed” and “re-crosslinked” at the same time, and at the threshold position, the proportion of collagen “re-crosslinking” is greater than the proportion of “destroyed”.

## 2. Materials and Methods

### 2.1. Vessel Tissue Welding Experiment

#### 2.1.1. H.F.E.W. Platform Set Up

In this section, we will discuss how to measure the temperature distribution of vessel tissue during high frequency welding using infrared observation and simulation. Figure 1 depicts an overview of the experimental setup.

First, we presented the experimental apparatus, associated factors, and the experimental technique of organizing a welding experiment. The infrared thermal imager was then utilized to examine the temperature distribution in the vessel tissue’s edge area throughout the experiment, while the COMSOL program was used to simulate the temperature distribution in the vessel tissue’s edge area and central area. Finally, the temperature observation data (from the infrared thermal imager) and temperature simulation data (from the COMSOL simulation) at the edge area were compared. When they were closely related, the simulated temperature distribution data could be deemed consistent with the actual temperature distribution data, and the simulated temperature distribution data in the central area could be used as the actual temperature distribution data.

The measurement method is based on two fundamental conditions: first, the temperature distribution data of vessel tissue is continuous, and there is no distortion in either time scale or space scale; second, the sample size of temperature observation data and temperature simulation data is large, excluding the influence of accidental factors.

#### 2.1.2. Vessel Tissue Preparation

Untreated ex-vivo porcine vessel tissue was obtained from a slaughterhouse and kept in 0.9% physiological saline. The experiment was completed in 12 h and all tissues were stored at 4 °C.

The vessel tissue was removed from the refrigerator, allowing it to reach room temperature (25 °C), then the blood vessel was cleaned in 0.9% physiological saline, the attached fat, muscle, and vessel branches were removed. The blood vessel was compressed with a pressure gauge at 30 N and its thickness was measured, which was 0.958 ± 0.122 mm. (all data presented as mean ± standard deviation). Vessel tissue was compressed under pressure without inelastic deformation or irreversible physical damage. To reduce the impact of dehydration on vessel tissue, vessel tissue was soaked in 0.9% physiological saline, welded in the open air, and then immersed in physiological saline again.

#### 2.1.3. Vessel Tissue Welding Process

EKVZ-300 was used for high-frequency welding in this experiment. An energy output controller and clinical surgical accessories are included in the system (power control pedal, hand-held laparoscopic forceps, etc.). The forceps apply pressure to fix the tissue while also transferring current to it via the electrode, facilitating tissue fusion through the synergistic action of electrothermal heating.

The prepared vessel tissue was placed on the lower-electrode and compressed (see Figure 2). The infrared thermal imager was adjusted directly above the square hole until both the welding edge and the center area were visible on the focal plane. The EKVZ-300, was started and the Cod-4 mode was selected for welding, and the infrared thermal imager began shooting; the data was transmitted to the IR software for recording; the equipment’s lead was connected to the oscilloscope (Pico 6403D 350 MHz Oscilloscope with AWG and Probes), and the electrical parameters were recorded by the voltage probe (Pico TA041 25 MHz, 700 V Differential Oscilloscope Probe) and current ring.

The welding time was 15–30 s and was divided into groups of 5 s.

#### 2.1.4. Temperature Measurement in Welding Process

The infrared thermal imager (Fortric227s) mounted on the insulating resin support records the temperature image of the edge area (non-electrode covered area) (see Figure 2) with time. Infrared temperature data was collected using an infrared thermal imager and the FOTRIC AnalyIR software. The sampling frequency was set to 30 Hz, and a total of 196,608 (384 × 512) data points per second were collected. This ensured that the system is able to record temperature changes during the welding process. The infrared lens uses M20-227s and the fixed focal length was 200 mm. Because there was no physical contact with biological tissue, the impact on the welding process could be ignored, and the temperature data accuracy was ±0.1 °C, the measurement system could measure the temperature more accurately and make the simulated environment closer to the actual operation.

### 2.2. COMSOL Simulation

Based on the welding experiment, a simulation model of container organization with the coupling of a biological heat transfer interface and electromagnetic heat interface was established. As shown in Figure 2, the upper and lower electrodes held the vessel tissue and were set as the excitation source and the grounding port. The vessel tissue was divided into several grid regions, and the temperature changes were solved and recorded. COMSOL Multiphsics 6.0 (Burlington, MA, USA) was used as the platform to make the simulation.

Pennes’ equation is a biological heat conduction equation that Pennes’ proposed in the 1940s [13]. At present, it is widely used in the fields of hyperthermia, laser therapy, radio-frequency therapy and so on.
(1)$$\rho c\frac{\partial T}{\partial t}=k\nabla^{2}T+q_{g}$$
where *ρ* is the tissue density (kg/m^3^), c is the tissue heat capacity (J/kg·K), *t* is the time (s), *k* is the tissue thermal conductivity (W/m·K), *T* is the tissue temperature (K), and *q*_g_ is the heat source (W) due to the externally induced electrosurgical heating of the tissue.

Using the temperature at the end of the previous time period as the starting point for the next time period, the temperature change of biological tissue can be regarded as a steady-state change, and the heat conduction equation of biological tissue at this time is:(2)$$\rho_{v}C_{p,v}u\cdot \nabla T_{2}+\nabla \cdot q=Q+Q_{bio}$$
(3)$$q=-k\nabla T_{2}$$
(4)$$Q_{bio}=\rho_{b}C_{p,b}\omega {\;}_{b}\left(T_{b}-T_{2}\right)+Q_{met}$$
where $\rho_{v}$ is the vessel density, $C_{p,v}$ is the vessel heat capacity at constant pressure, $T_{2}$ is the vessel tissue temperature (K), $Q_{bio}\;$is the biological heat source, $\rho_{b}$ is the blood density (kg/m^3^), $C_{p,b}$ is the blood specific heat capacity (J/kg·K), $\omega_{b}$ is the blood perfusion rate(1/s), $T_{b}$ is the arterial blood temperature (K) and $Q_{met}$ is the metabolic heat source (W/m^3^). Because the experiment was carried out in vitro, $Q_{bio}$ was omitted in the calculation, and the effects of blood perfusion and body heat production on temperature were not considered.

Except for tissue density, which can be considered constant throughout the experiment, other parameters will change with changes in temperature or electrical parameters. The tissue of the vessel is made up of 70% water and 30% solid matter [14]. Tissue warming will undoubtedly reduce tissue water content and alter tissue thermal conductivity. According to Chen R. K. [12], the relationship between thermal conductivity and temperature can be described as:(5)$$k_{v}=k_{0}+0.0013\left(T_{2}-T_{0}\right)$$
where *k*_0_ (0.462 W/m·K) is the baseline tissue thermal conductivity at *T*_0_ (25 °C) [15].

The conductivity of biological tissue is not only affected by temperature but also related to current frequency [15]. Haemmerich D. [16] uses tissue conductivity σ reduced to one ten thousandth of the original to indicate thermal damage. Chen R. K. [12] represented the relationship between conductivity and temperature as:(6)$$\sigma_{v}=\sigma_{0}\left[1+0.02\left(T_{2}-T_{0}\right)\right]$$
where *σ*_0_ is the reference electrical conductivity of the tissue (0.24567 S/m, see Table 1) and *T*_0_ is the reference temperature (25 °C).

The basic properties of electrode materials can be considered constants throughout the experiment and are listed in Table 2.

### 2.3. Vessel Tissue Welding Strength Test

When H.F.E.W. is used for intestinal or small-size blood vessel welding, bursting pressure is typically used as the strength inspection index, which is the most commonly used strength inspection method [18,19], but it is difficult to determine bonding mechanics through it. The maximum external force that the welding area can withstand in the actual strength test is related not only to strength but also to the shape and structure of the blood vessel. Some researchers have tried to test the welding strength by stretching or shearing [20,21], but there is a problem: the unclear definition of the welding area leads to great differences between the welding strength data of the same biological tissue obtained by different researchers. For example, for the welding area with the shape of 20 × 20 mm and the welding area with the shape of 10 × 2 mm, even if they are put into the same high-frequency electric field for high-frequency welding with the same time length, the welding strength is obviously different, but researchers seem to have avoided this problem. Cezo J.D. [22] employed the T-peel test (vessel tissue was stretched into a T shape during the test, see Figure 3) to provide a more detailed evaluation standard for welding strength analysis. On this basis, we improved the processing method of welding strength data. Considering the difference in shape and size of the welding area, we divided the welding strength by the width of the welding area to obtain the strength of the welding area per unit width.

The welding vessel tissue was cut along the axis to the edge of the welding area (do not damage the welding area) and loaded on the upper and lower jaw of the tension meter (see Figure 3), and the vessel tissue was stretched at the speed of 2.00 mm/s until it was broken.

Different vessel tissues were grouped according to the welding time, and the welding strength of all samples in each group was recorded (*n* = 10), the maximum and minimum values, were marked and the average strength of each group of samples was calculated. The welding strength of the samples with no welding area or “damaged” welding area was not recorded.

### 2.4. Raman Spectrum Test

Raman spectroscopy can provide information about the secondary structure of proteins and is widely used to analyze the conformation of proteins, especially the structural comparison of collagen before and after thermal denaturation. Alimova A. [9] uses the strength rate of amide Ⅲ band (1247 cm^−1^), which is mostly from the collagen to 1325 cm^−1^ band (the superposition of elastin and keratin band) to measure the denatured healing process of collagen in skin tissue. This method can also be used to evaluate the effect of temperature distribution on the physicochemical properties of collagen in vessel tissue.

Raman spectrometer (NTEGRA Spectra-Ⅱ + In Via Qontor) was used to analyze the central area of vessel tissue (excitation wavelength: 785 nm). The central area of the vessel tissue sample was removed and rinsed with deionized water to remove the attached damaged adipose tissue (reddish brown residue at this time). The Raman spectrum was collected from the central area, and multiple frequency points were collected for each sample, and the average value was taken to reduce the influence of signal noise.

The Raman spectrum data obtained were edited by Labspec software and smoothed at first, then the baseline is removed by polynomial fitting, and the range of the Raman spectrum was limited to 800–2000 cm^−1^. Finally, the Gauss–Loren function was used to calibrate the characteristic peak, and the information of characteristic peak position, extreme value, FWHM and so on were derived.

## 3. Results

### 3.1. Vessel Tissue Welding Result 

Welding experiments lasted 10 s, 15 s, 20 s, 25 s, and 30 s. The vessel tissue failed to show obvious morphological changes on the surface in the 10 s and 15 s grouping experiment (at this time, it is difficult to distinguish the morphological changes in the welding center from the changes in physical properties caused by an external electric field). The residues on the surface of the vessel transformed into glass-like tissue (See Figure 4) and can be seen directly in the groups of 20 s, 25 s, and 30 s (at this time, the light transmittance of the welding area increases and the thickness decreases significantly).

### 3.2. Temperature Measurement and COMSOL Simulation Result

The temperature was transferred outward from the organization’s central area during the welding process (and then heat exchange with air or metal electrode). There was a clear gradient change in the *x*-axis direction and a subtle gradient change in the *y*-axis direction. The main issue is whether there is an obvious temperature change in the *z*-axis direction, which will have a direct impact on the accuracy of the temperature data. The shape of the isothermal surface obtained by COMSOL simulation can be seen by intercepting the ellipsoid by the cube space. There is a temperature difference along the *z*-axis (see Figure 5).

As the heat was transferred along the *x*-axis, the temperature difference in the *z*-axis grows larger and larger. The deviation of the isothermal surface reaches 0.24 mm in the outermost region, and the deviation rate is 5.6% when compared to the width of the thermal diffusion region (4.32 mm). At this point, the isothermal surface is roughly defined as a plane perpendicular to the *x*-*z* plane, the internal temperature of the tissue is roughly equal to the surface temperature, and there is no temperature difference in the *z*-axis direction.

When entering electrical parameters into COMSOL, we chose to directly import the voltage, frequency, duty cycle, and other data stored in the oscilloscope, and then modify the waveform by defining an analytical formula to make it a standard square wave waveform, to reduce interference from noise and other factors during measurement. When temperature data from an infrared thermal imager was unevenly distributed, the primary cause was that residual adipose tissue attached to the surface or the cavity left by the removal of blood vessel branches affect the conduction or distribution of surface temperature. Second, when thermal denaturation occurred in the central area of vessel tissue, the vessel tissue in the marginal area curls up, affecting the validity of the observed data; however, this influence can be avoided by expanding the area of vessel tissue and fixing the vessel tissue on the electrode base.

Figure 6 shows the temperature change curve of the edge area under different welding times. With the increase of distance (*x*-axis), the simulated temperature curve gradually decreases, while the measured temperature curve first rises and then falls. There is a deviation between the data measured by the infrared thermal imager and the data obtained by simulation in the range of about 1mm from the edge of the electrode, and then the two temperature curves are highly fitted. The reason for the deviation is that, in the experiment, vessel tissue gradually warms up under the action of electrothermal coupling, and the heat flows from the high temperature area (vessel tissue) to the low temperature area (metal electrode), the electrode absorbed heat. As a result, the highest temperature in the edge area of the vessel is not close to the electrode but shifts outward for a distance. The fitting degree of the two curves (simulation data from edge area and infrared thermal data) in Figure 6d, Figure 6e, and Figure 6f is 98.12%, 97.25% and 98.50%, respectively.

In the central area covered by electrodes, it is impossible to observe the temperature distribution with an infrared thermal imager. The comparison of the above experimental results shows that the COMSOL simulation edge area data accurately reflects the temperature distribution during vessel welding. Therefore, we can use this model to calculate the temperature distribution in the welding process in the central area.

As simulation (central area) data is shown in Figure 6d–f, the vessel tissue temperature does not obviously change with the change in distance. This is because the heat transfer in the central area only occurs between biological tissues. Compared with electrode materials, as shown in Table 1 and Table 2, vessel tissue has larger heat capacity and smaller thermal conductivity, so the temperature distribution is more balanced. When the welding time is 20 s, the central area’s temperature is about 50 °C and the highest temperature is 53.6 °C (see Figure 6g). When the welding temperature is 25 s, the central temperature is about 58 °C, and the highest temperature is 62.5 °C (see Figure 6h). When the welding time is 30 s, the central temperature is about 66 °C, and the highest temperature is 72.2 °C (see Figure 6i).

### 3.3. Vessel Tissue Welding Strength Test Result

The welding area’s width changes slightly during the T-peel test because the welding structure is sheared. To control the error and meet the experimental criterion of “single variable“, the welding strength data were normalized and expressed as F-unit, which is the external force per unit width, in N/mm.

The welding strength increases as the welding time increases see Figure 7 and Table 3. The welding strength data for each group with a 5 s interval differ significantly. The average change rate slows down near the time point of 25 s, but the data with a welding time of 30 s clearly does not conform to this law when compared to the previous group (the welding time is 25 s). Welding strength increased by 116.99% on average. At this time, however, the vessel tissue was severely damaged by heat. As shown in Figure 5, heat accumulated near the edge of the electrode, which led to the breakage of some samples.

### 3.4. Raman Spectrum Result

The processed Raman spectrum data are shown in Figure 8, and the images are grouped at 5 s intervals.

In Table 4, the characteristic peak at the 1247 cm^−1^ (β-sheet) [23] position of the amide Ⅲ band shifted red, and when the welding time was 30 s, the peak position shifted to 1266 cm^−1^ (α-helix). This means that the β-sheet inter-chain hydrogen bonds are destroyed under the action of the high-frequency electric field, and one part of them is transformed into a closed loop composed of hydrogen bonds, the β-sheet is transformed into α-helix, and the other part is disordered [24]. The characteristic peak at the 1325 cm^−1^ position of amide Ⅲ band also shifted to blue, and the peak position shifted to the 1305 cm^−1^ position, The reason is the formation of hydrogen bonds in α-helix structures makes the corresponding amide Ⅲ band coordinate shift to a low wavenumber [25].

The half-width of the Raman characteristic peak is affected by the purity of the sample to be detected. When the peptide chain changes from an ordered structure to a disordered curl, the Raman spectrum gradually widens, and the half-width of the characteristic peak increases accordingly. Otherwise, the half-width decreases [26,27].

When the welding time is 15 s and 20 s, the peak value of I1247 cm^−1^ is less than that of I1325 cm^−1^ (the characteristic peak position will shift due to the influence of stress, pressure, temperature, deformation and other factors), but at 30 s, the peak value of I1247 cm^−1^ is much lower than that of I1325 cm^−1^.

## 4. Discussion

A combination of COMSOL simulation and infrared thermal image observation was used in this research to indirectly provide the temperature distribution of the vessel tissue during high-frequency welding by comparing temperature data of the welding tissue’s edge area.

In the research of collagen peptides, it was discovered that differences in repeat units of collagen peptides resulted in significant changes in thermal stability, as shown in Table 5.

Paton B.E. [1] believes that welding strength is caused by the thermal denaturation of unstable globular proteins, followed by structural transformation, and finally colloidal material formation. Kramer E.A. [8] believes that when exposed to energy at 40–60 °C, helical collagen denatured into amorphous peptide chain curls composed of three helices. Therefore, we can assume that in the welding process, with the increase in energy input and temperature, the collagen in the tissue goes through the process of “cross-linking, unspinning and re-cross-linking”. The starting point for collagen denaturation intensity was set at 40 °C, and the starting point for collagen thermal damage was set at 60 °C. The following is a description of each phase:

### 4.1. Phase I—Foundation Temperature Rise (Initial Temperature to 40°C)

The initial temperature of the vessel tissue is 25.0 °C after measurement, and the data simulated by COMSOL show that when the welding lasts for 11.5 s (see Figure 9), the temperature in the welding center area reaches 40 °C, the heat diffusion area is about 4 mm wide on both sides of the center area, diffuses from the electrode contact area to both sides, the temperature decreases from 35.4 °C to 25.0 °C, the cooling rate decreases from high to low, and the temperature change curve decreases smoothly; this is also fitted with the data directly observed by the infrared thermal imager.

In Phase I, when the welding time is 10 s, the strength is 0.097 N/mm. It is worth mentioning that some of the samples do not even form a welding area and can not be tested.

### 4.2. Phase II—Denaturation (40–60 °C)

The temperature of the vessel tissue is still rising in phase II. The temperature in the central area reaches 60.0 °C at 23.5 s (see Figure 9), according to COMSOL simulation data. Similar to phase I, the tissue temperature decreases from the center to the periphery, the width of the heat diffusion area remains about 4 mm, and the temperature drops from 53.2 °C to 25.0 °C, owing to biological tissues’ higher heat capacity and lower thermal conductivity, which prevents uniform heat conduction in a short period of time.

The collagen peptides in collagen molecules have reached the temperature threshold that can cause thermal denaturation, according to the research findings [29,30,31,32,33]. According to the data of Raman spectra, shown in Figure 9, when the welding time is about 15 s, the I1247 cm^−1^/I1325 cm^−1^ rate is 2.2927, and the welding strength is 0.154. When the welding time is 20 s, the rate of I1247 cm^−1^/I1325 cm^−1^ drops to 1.6714, and the corresponding welding strength is 0.244 N/mm, which indicates that collagen is undergoing denaturation, interchain hydrogen bond breakage, and becoming amorphous peptide curls. New cross-molecular links are also being formed. The central area of vessel tissue began to shrink macroscopically, the thickness gradually decreased, and the color changed from light pink to dark. The area with adipose tissue on the surface is left with a dark brown residue. In the thoroughly cleaned vessel area, which is similar to the “membrane”, there is increased light transmittance (see Figure 4).

### 4.3. Phase III—Thermal Damage (60 °C—Maximum Temperature)

Because the vessel tissue had been “burned out” during the 30 s welding experiment, 30 s was chosen as the endpoint of the welding time. According to COMSOL simulation data, the maximum temperature in the central area is 72.4 °C at this time (see Figure 9), and the width of the thermal diffusion area is about 4 mm. The temperature drops from 64.2 °C to 25.0 °C.

The positive correlation between temperature rise and welding duration changes in Phase III. The tissue strength reaches the theoretical maximum as collagen is fully denatured by the synergistic action of current and heat and forms a transparent and solid weld, but the continuously accumulated heat will also damage the tissue in the non-welding area and even “cut” the welded vessel tissue like a scalpel.

As shown in Figure 9, the rate of I1247 cm^−1^/I1325 cm^−1^ increases to 1.8351 when the welding time is 25 s, which indicates that under the electrothermal synergistic effect of collagen, the amorphous peptide chain is re-crosslinked to form a network structure, making the vascular tissue weld together. At this time, the welding strength is 0.321 N/mm. A glassy band appears in the center of the welding area, with increased light transmittance and significantly reduced thickness (see Figure 9).

As the heat continues to accumulate in the tissue, the effect of heat damage becomes more and more apparent as collagen is re-cross-linked, even over and above cross-linking. When the welding time was 30 s, the rate of I1247 cm^−1^/I1325 cm^−1^ dropped sharply to 0.7983, indicating that the collagen network structure generated by re-cross-linking was damaged due to the accumulated heat. As shown in Figure 9, the welding area is completely transformed into a glassy structure, at which the welding strength reaches 0.697 N/mm. However, it may also be shown in Figure 9 that the central area is burned by accumulated heat until it breaks; although certain welding strength can still be maintained on both sides, it is not recorded.

As can be seen from Figure 9, the Funit (strength) curve obtained by T-peel increases monotonically, but considering that the success rate for “30 s” is less than 100%, the maximum value of the Funit (strength) curve should be “25 s”. The point of local maximum of the rate curve of I1247 cm^−1^/I1325 cm^−1^ obtained by the Raman spectrum was also at “25 s”, when the proportion of collagen re-cross-linking was the largest. Considering the strength and thermal damage, when the welding time was set to 25 s, better welding strength data could be obtained under the condition of controlling thermal damage.

According to the data in Table 5 and the researchers’ experiments [8], we divided the welding process into three phases. In Phase I, the collagen cannot be fully denatured to form a firm welding area. In Phase II, the peptide chain in collagen is gradually denatured, and at the same time, “untwisting and re-crosslinking” occurs. In Phase III, the vessel tissue may produce higher welding strength or be “burned out”. So we should control the welding process to Phase II and stop before Phase III. The corresponding temperature range is 40–60 °C.

According to the welding strength data (as shown in Figure 7) detected by the T-peel test, the welding strength increases with the extension of welding time. However, considering that when the welding time is 30 s, the vessel tissue is likely to be “burned out”, on the premise of ensuring safety, the welding time should be around 25 s, greater than 20 s and less than 30 s, and the corresponding temperature should be 53–71 °C.

According to the Raman spectrum data (as shown in Figure 8), when the welding time is 15–20 s, the “cross-linking” ratio in collagen decreases with time, when the welding time is 20–25 s, the “cross-linking” ratio in collagen increases with time, and when the welding time is 25–30 s, the “cross-linking” ratio in collagen decreases with time. The welding time should be controlled in 20–25 s, and the corresponding temperature should be 53–62 °C.

The intersection of the three temperature ranges obtained by the above analysis shows that the best temperature range in this experiment may be 53–60 °C.

## 5. Conclusions

In this study, an observation method combining optical measurement and simulation was proposed to study the temperature distribution of vascular tissue during welding. This method has the advantages of a large sample size, high fitting degree (>97%) and wide application range. Through strength tests and Raman spectrum analysis, it is found that the influence of temperature distribution on vascular tissue during welding is not monotonically increased or decreased, but that the optimum welding conditions maybe exist within the temperature range (53–60 °C). At present, this method has been tried to be applied in intestinal welding experiments and will be further expanded in the future to help researchers explore the optimal welding parameters in clinical welding operations.

In the future, there are several improvement directions as follows: First, simplify the experimental device, fix the micro camera on the surgical forceps, and transmit the temperature distribution image collected by optical fiber to the back-end workstation, so as to increase the practicability of this technology. Secondly, the temperature simulation models of different tissues were established, the temperature data collected were compared in real time, and the optimal welding parameters were fed back to the medical workers. Finally, the temperature of the forceps was recorded by micro-thermocouple to avoid the deviation of the temperature distribution curve and increase the degree of data fitting.

## Figures and Tables

**Figure 1 biosensors-12-00209-f001:**
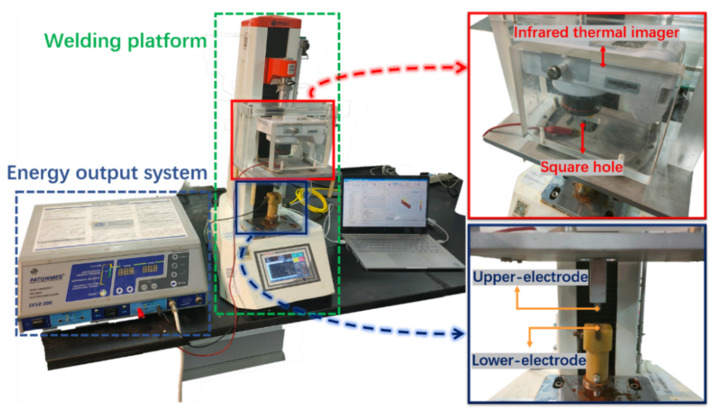
Physical map of H.F.E.W. experimental setup. In the experiment, the rigid structure composed of a tensimeter base, infrared thermal imager and upper-electrode compresses the tissue downward, the infrared thermal imager can record the temperature distribution in the welding process through the square hole, and the electrical parameters are measured and displayed by an oscilloscope (including current ring and voltage probe) (not shown).

**Figure 2 biosensors-12-00209-f002:**
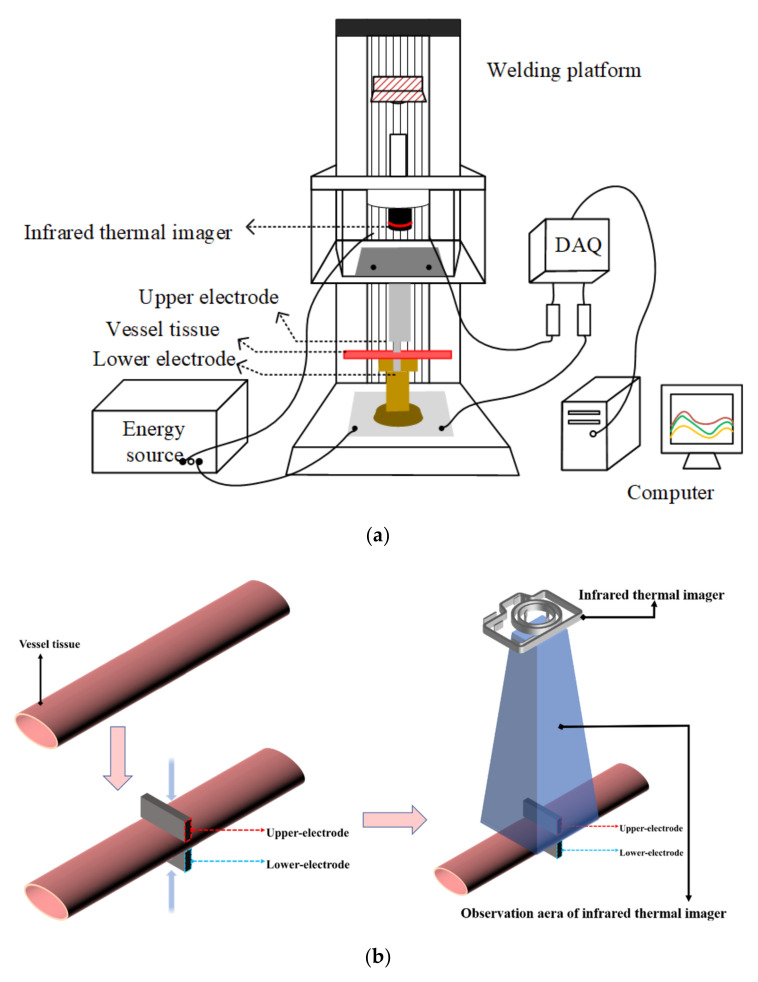
The treated vessel tissue is placed between the upper and lower electrodes, and sufficient pressure is applied to fix it. After the pressure is stabilized, the upper and lower electrodes deliver energy to the vessel tissue, and the infrared thermal imager begins to record the tissue temperature until the end of the welding experiment. (**a**): Schematic diagram of H.F.E.W. experimental setup. (**b**): Flow chart of vessel tissue welding experiment.

**Figure 3 biosensors-12-00209-f003:**
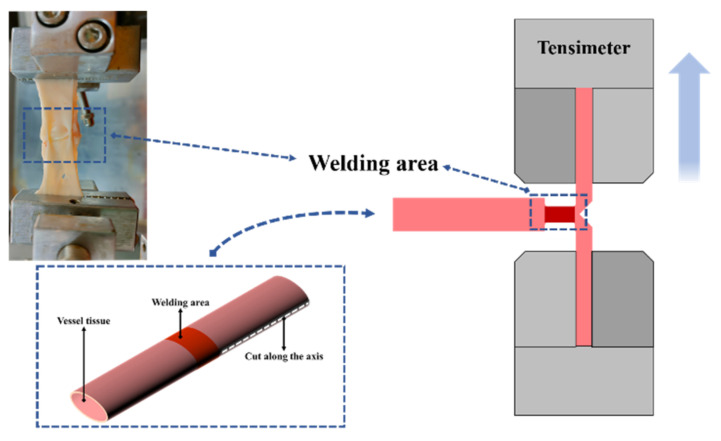
Simplified schematic diagram of T-peel test. Cut one end of the vessel tissue along the long axis of the vessel to the edge of the welding area (as shown in the lower left corner) so that the vessel tissue at this end changes from a complete wall structure to two lobes of vessel tissue attached to the welding area. The two cleaved vessel tissues are loaded into the upper and lower jaws of tensimeter.

**Figure 4 biosensors-12-00209-f004:**
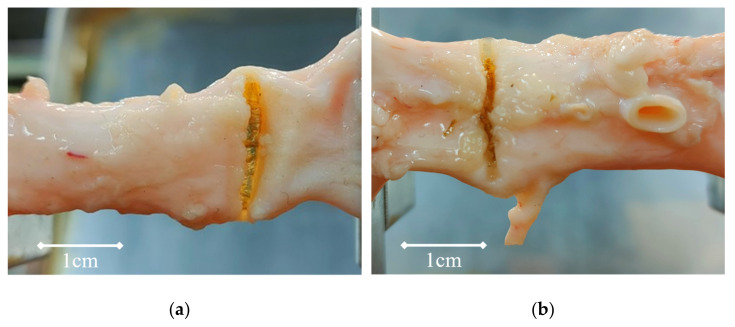
(**a**): When the welding time is 30 s, the welding area becomes glass-like tissue, the thickness decreases and the light transmittance increases. (**b**): When the welding time is 20 s, the welding area changes from pink to translucent white, and the reddish-brown floc is the thermal denaturation product of fat.

**Figure 5 biosensors-12-00209-f005:**
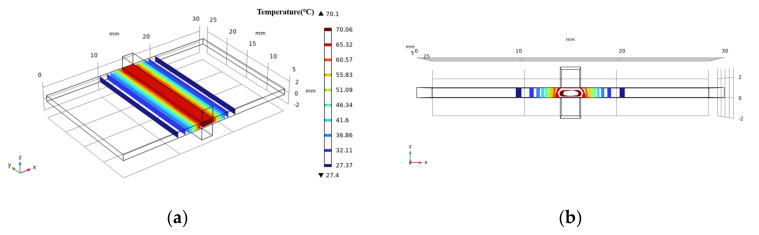
Simulation diagram of medium temperature surface during experiment (COMSOL). The section is part of an ellipsoid on (**a**). In the *z*-axis direction, there is a certain temperature difference inside and outside the tissue, and the isothermal surface deviates a little on (**b**).

**Figure 6 biosensors-12-00209-f006:**
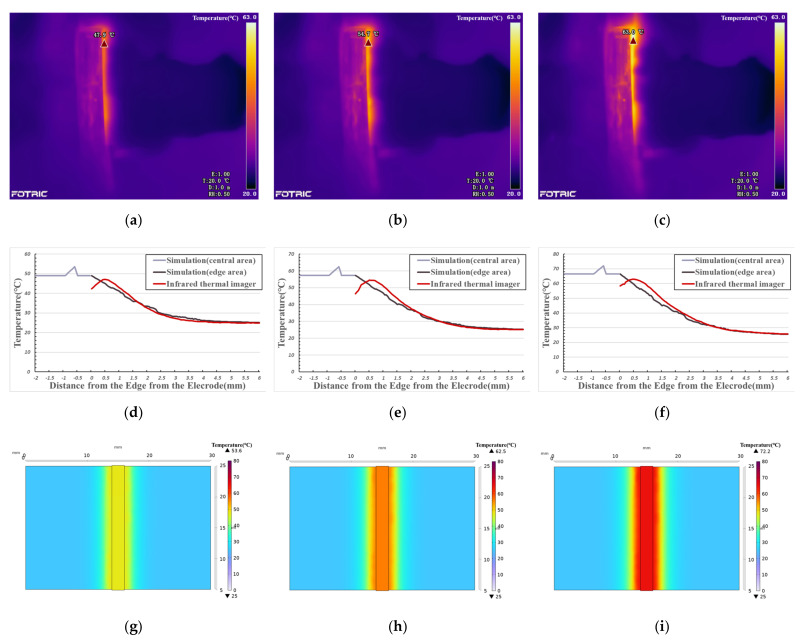
(**a**): Thermal image (20 s); (**b**): Thermal image (25 s); (**c**): Thermal image (30 s); (**d**–**f**): Temperature comparison curve between infrared thermal imager data and simulation data; (**d**): 20 s; (**e**) 25 s; (**f**): 30 s; (**g**–**i**): Simulation image of temperature distribution with the same visual angle as thermal image. (**g**): corresponds to (**a**), 20 s; (**h**): corresponds to (**b**), 25 s; (**i**): corresponds to (**c**), 30 s.

**Figure 7 biosensors-12-00209-f007:**
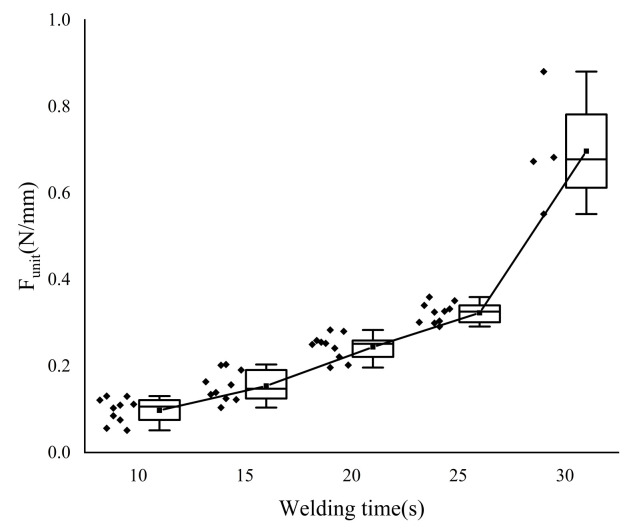
T-peel test results. In order to calculate the average change rate when the welding time is 15 s in Table 3, we added additional welding strength data (welding time is 10 s) in the figure. Scattering points on the left side of the box are specific sample data points. Each group has 10 sample data. However, when the welding time is 30 s, some vessel samples are “burned out”, so there are only four samples.

**Figure 8 biosensors-12-00209-f008:**
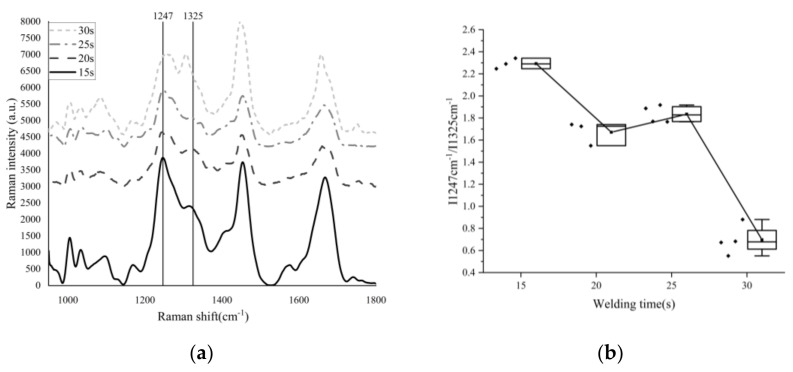
(**a**): Raman spectrum data of vessel tissue with different welding times; (**b**): Density rate of I1247 cm^−1^ and I1325 cm^−1^. Scattering points on the left side of the box are specific sample data points. *Y*-axis coordinates only represent the Raman intensity of “15 s”. We vertically shift the Raman spectrum image of “20 s”,”25 s”and”30 s” in the *Y*-axis direction to different degrees, in order to compare the differences between them conveniently.

**Figure 9 biosensors-12-00209-f009:**
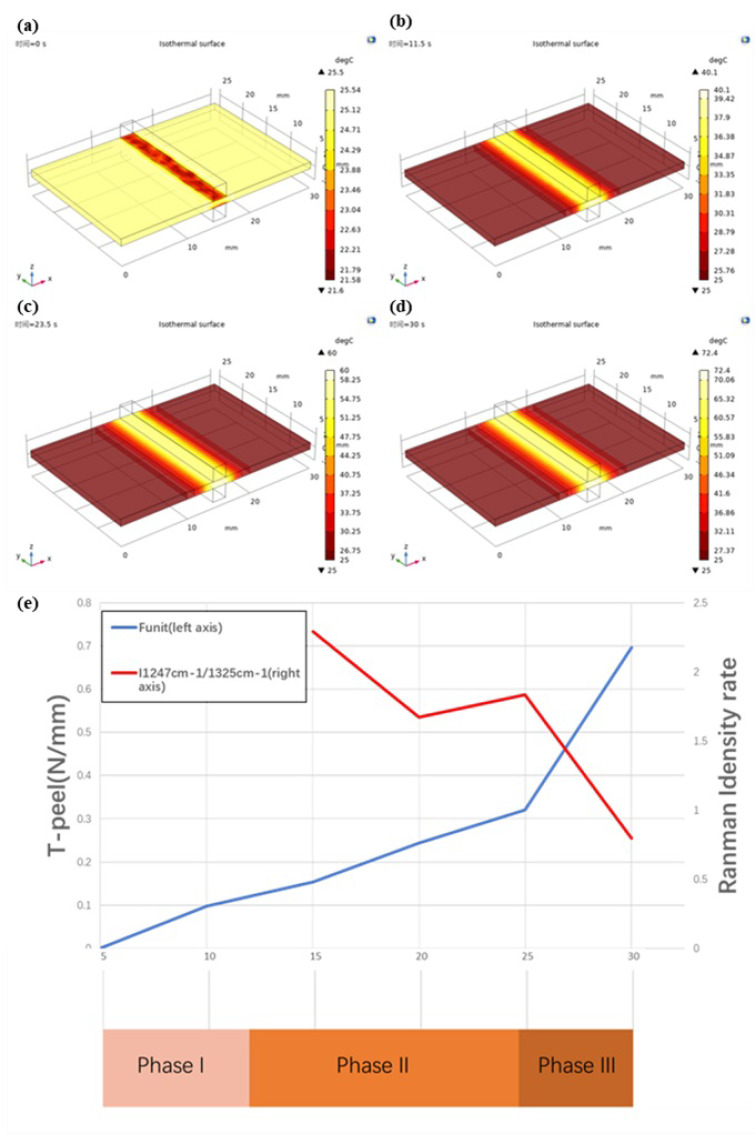
T-peel data and Raman Spectrum data at different phases. (**a**): At the beginning of the welding experiment(t = 0 s), the initial temperature of vessel tissue is about 25 °C The central area of the vessel is fixed by a low−temperature (about 20 °C) metal electrode, and the temperature is reduced. (**b**): When the welding time is 11.5 s, the temperature in the central area of vessel tissue reaches 40 °C and spreads to both sides. (**c**): When the welding time is 23.5 s, the temperature in the central area of vessel tissue reaches 60 °C and spreads to both sides. (**d**): When the welding time is 30 s, the highest temperature of vessel tissue reaches 72.4 °C. (**e**): The left axis coordinates of the line chart represent the T−peel test strength, the right axis represents the ratio of I1247 cm^−^^1^/I1325 cm^−^^1^ and the horizontal axis coordinates represent the welding time. Phase I: 0−11.5 s, Phase II: 11.5−23.5 s, Phase III: 23.5−30 s.

**Table 1 biosensors-12-00209-t001:** The basic properties of the vessel.

Property	Expression	Unit	Reference
Density ($\rho_{v}$)	1101.5	kg/m^3^	[15]
Thermal conductivity ($k_{0}$)	0.462	W/m·K	[15]
Heat capacity at constant pressure ($C_{p,v}$)	3306	J/kg·K	[15]
Electrical conductivity ($\sigma_{0}$)	0.24567	S/m	[15]

**Table 2 biosensors-12-00209-t002:** The basic properties of electrode materials.

Property	Expression	Unit	Reference
Density ($\rho_{m}$)	8960	kg/m^3^	[17]
Thermal conductivity ($k_{m}$)	401	W/m·K	[17]
Heat capacity at constant pressure ($C_{p,m}$)	381.875	J/kg·K	[17]
Electrical conductivity ($\sigma_{m}$)	57,142,857	S/m	[17]

**Table 3 biosensors-12-00209-t003:** T-peel test result.

Welding Times (s)	Strength (N/mm)			Average Change Rate (%)
	Max	Min	Average	
10	0.13	0.05	0.10	-
15	0.20	0.10	0.15	58.34%
20	0.28	0.20	0.24	58.51%
25	0.36	0.29	0.32	31.57%
30	0.88	0.55	0.70	116.99%

**Table 4 biosensors-12-00209-t004:** Raman Spectrum Result.

Welding Times (s)	Position (cm^−1^)	Intensity	Half-Width (cm^−1^)	I1247 cm^−1^/I1326 cm^−1^
15	1247.30	3227.29	37.00	2.29
	1322.71	1409.34	51.15	
	1246.21	3220.33	41.38	2.25
	1315.23	1433.47	72.50	
	1247.30	3038.43	41.37	2.34
	1312.01	1297.59	74.68	
20	1247.30	1087.83	52.28	1.72
	1317.37	630.84	95.95	
	1250.56	1856.38	56.61	1.74
	1313.08	1158.75	129.73	
	1249.47	1357.60	54.44	1.55
	1306.66	876.44	110.86	
25	1248.38	1704.50	43.54	1.92
	1319.51	888.80	76.68	
	1245.13	2165.38	39.20	1.77
	1315.23	1223.91	87.36	
	1246.21	2669.64	43.56	1.77
	1322.71	1511.89	78.74	
	1249.47	1464.48	43.53	1.89
	1314.16	775.89	85.26	
30	1266.29	3484.08	75.66	0.68
	1303.85	5100.98	38.42	
	1261.98	1086.55	71.38	0.72
	1304.92	1502.61	38.41	
	1255.51	422.15	47.59	0.67
	1310.26	634.66	31.98	
	1258.15	3203.23	97.10	0.99
	1307.73	3251.95	95.99	
	1253.91	1869.72	60.94	0.94
	1304.51	1999.29	62.02	

**Table 5 biosensors-12-00209-t005:** Thermal denaturation data of collagen peptides.

Number	${\left(X-Y-Gly\right)}_{n}$	T_d_ (°C)	$\Delta H_{vH}^{0}/kJ\cdot {\mathrm{mol}}^{-1}t$	References
1	${\left(\mathrm{Pro}-\mathrm{Hyp}-\mathrm{Gly}\right)}_{7}$	36	-	[28]
2	${\left(\mathrm{Pro}-\mathrm{Pro}-\mathrm{Gly}\right)}_{7}$	-	-	[28]
3	${\left(\mathrm{Hyp}-\mathrm{Hyp}-\mathrm{Gly}\right)}_{10}$	65	-	[28]
4	${\left(\mathrm{Pro}-\mathrm{Hyp}-\mathrm{Gly}\right)}_{10}$	61–69	-	[28]
5	${\left(\mathrm{Pro}-\mathrm{Pro}-\mathrm{Gly}\right)}_{10}$	25	−18.8	[29]
6	${\left(\mathrm{Pro}-\mathrm{Pro}-\mathrm{Gly}\right)}_{10}$	34	−18.6	[30]
7	${\left(\mathrm{Pro}-\mathrm{Pro}-\mathrm{Gly}\right)}_{10}$	31–41	-	[31,32]
8	${\left(\mathrm{Pro}-\mathrm{Hyp}-\mathrm{Gly}\right)}_{10},\mathrm{pH}1$	60.8	−23.6	[33]
9	${\left(\mathrm{Pro}-\mathrm{Hyp}-\mathrm{Gly}\right)}_{10},\mathrm{pH}7$	57.8	−23.3	[33]
10	${\left(\mathrm{Pro}-\mathrm{Hyp}-\mathrm{Gly}\right)}_{10},\mathrm{pH}13$	60.8	−25.1	[33]

## Data Availability

The datasets generated during and/or analyzed during the current study are available from the corresponding author on reasonable request.

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
