# Peer review of "Temperature Distribution of Vessel Tissue by High Frequency Electric Welding with Combination Optical Measure and Simulation"

_biosensors, 2022, doi:10.3390/bios12040209_

Round 1

Reviewer 1 Report

In this article, a method combining optical measurement and simulation was presented to evaluate the temperature distribution of vascular tissue during high frequency electric welding. The experimental proceeding and COMSOL simulation are demonstrated specifically in details, which I have no doubt about your results. The only question is why the shape of the electrodes you used to simulate and applied to the vascular tissue was two rectangle blocks, without blades? In my opinion, the blade can help to increase the intensity of electric field locally and make the cut sharply. So I hope the authors can consider my suggestion and optimize your work.

Reviewer 2 Report

The paper by Wang et al describes electric welding of vessels and estimation of such process with temperature measurements combined with mechanical and optical tissues studies. The theme of the paper maybe quite interesting to a wide range of readers, at the same time there are numeral drawbacks that must be corrected prior to publication.

Major issues:

  1. In the end of the Introduction section (formulation of the study aim) the authors only propose to improve the temperature measurements during welding. At the same time, there must be mentioned other tissues estimations (mechanical and optical) performed during the study.
  2. Figure 1 demonstrates only a picture of utilized setup, but the authors must explain the working principle of such setup. Please, provide schematic explanation of the setup.
  3. Section 2.1.2. What type of vessels was utilized? One may assume that the vessel tissues were obtained from soma animals, please, specify.
  4. Why does the authors used welding time from 15s to 30s?
  5. Add to table 1 and table 2 appropriate references to support the proposed values.
  6. Please, add wide explanation to figure’s legends; eg in figure 3 legend there is no any explanations to the performed test (I can’t see a T-shape within the picture of vessel). The same question for figures 7 and 9.
  7. Section 2.4 is missing information regarding laser characteristics. What was the diameter of laser spot? What was the laser intensity on the vessel tissue? Time of laser exposure? Maybe the laser radiation itself caused some changes in vessel tissue? In addition, there are some strange phrases as “The Raman spectrum was collected from the central area, and multiple frequency points are collected for each sample, and the average value is taken to reduce the influence of signal noise.”, so how many points were measured? How the exact point of measurement (region of interest) was chosen? Maybe depict such information in the photograph of melted vessel (figure 4)?
  8. Figure 5b is missing dimensions and there is no color bar for entire figure 5.
  9. Figure 6g, h, e is not properly explained. Where these temperature distributions were acquired?
  10. Figure 7 in section 3.3 is missing information regarding the number of analyzed samples. How many tests were performed? Add such information as dots to the plotted whisker and box diagram, and add such information to materials and methods.
  11. The authors stated materials and methods that welding time was varied from 15 to 30 s, but figure 7 contains data for 10s.
  12. What is shown in table 3? What means min and max? How the average change rate was calculated?
  13. The number of significant digits in tables 3 and 4 is quite strange.
  14. Figure 8b: add dots (that represent a single measurement) to whisker and box diagram.
  15. Figure 9 is quite complex, but the figure legend is missing any explanations
  16. Lines 396-398: “The maximum value of the rate curve of I1247 cm-1/I1325 cm-1 obtained by Raman spectrum was also at “25 s”, when the proportion of collagen re-cross-linking was the largest.” But the figure 9 demonstrates that the maximum rate is at 15s. So the entire paragraph is confusing.
  17. In my opinion, the statement in the conclusions “This method has the advantages of large sample size, high fitting degree (>98%) and wide application range.” is not supported by results. Moreover, in the abstract the authors denote 97.4%, the main text is missing calculations and in the abstract it is >98%.
  18. The statement in the conclusions “…but that the optimum welding conditions exist within the temperature range.” is not supported by the data. May the authors debate?
  19. The number of references is too small. Many parts of the text require supporting references (eg section 3.3).

Minor Issues:

  1. Line 37 – double dots in the end of the sentence.
  2. In reference to previous studies use similar references (as sometimes the authors refer as “Yang CH…” and sometimes “A. Alimova…”).
  3. Figure 4 is missing dimensions.
  4. Figure 6 d,e,f: both simulations are shown in blue color, please, add color or it is impossible to distinguish edge area and central area.
  5. Figure 8a is missing y-axis legend.

In general, in present form the paper is missing important parts that may prove the highlighted conclusions. The authors must clear the mentioned issues before the publication.

Reviewer 3 Report

This manuscript deals with a biomedical observation method  to study the temperature distribution of vascular tissueduring welding (combining optical analysis and simulation). As such, this topic is not related to Biosensors or Biosensing, so it is recommended to be transferred to a journal devoted to biomedical research.

Reviewer 4 Report

The temperature distribution is presented in Figure 5, but the temperature values are not shown. Please add Temperature values.
 Figure 6 has many disadvantages: Signatures are very small and unreadable. Figures g), h) and i) looks the same.

Author Response

请参阅附件。

Round 2

Reviewer 2 Report

I like the way how the authors improved the manuscript, now it looks really easy for readers to understand what was made by the authors.

Only a couple of suggestions:

  1. Figures 5, 6 etc now have color barsb, but these bars missing the dimension. So add T sign to highlight that temeperature is presented.
  2. Despite Alimova and even prof. Alfano missed y-axis exact explanation in their figures, I believe that the authors must add y-axis legend as "Raman intensity (a.u.)" and explain in figures legend that Raman spectra are shifted by y-axis to clearly show the differences in the spectra.
